# Decoupling Economic Growth from Fossil Fuel Use—Evidence from 141 Countries in the 25-Year Perspective

**Katarzyna Frodyma** *[ID], **Monika Papież** [ID] **and Sławomir Śmiech**

Department of Statistics, Cracow University of Economics, Rakowicka 27, 31-510 Cracow, Poland;
papiezm@uek.krakow.pl (M.P.); smiechs@uek.krakow.pl (S.Ś.)
* Correspondence: frodymak@uek.krakow.pl; Tel.: +48-12-293-5205

**Abstract:** This study offers an in-depth analysis of the decoupling of economic growth from fossil fuel use in 141 countries over the last 25 years. The study is based on the Tapio decoupling approach, and two methods of measuring fossil fuel use, i.e., domestic material consumption (DMC) and material footprint (MF), are applied. Groups of countries with similar decoupling patterns are identified through the k-medoids method. Next, the relationship between these patterns and the level of countries' development is examined. The results reveal that using different measures of fossil fuel use yields different processes of decoupling economic growth from fossil fuel use. In particular, when the DMC indicator is considered, relative decoupling is observed in most analysed cases. When the MF indicator is applied, the decoupling states of individual countries change more frequently. Finally, in highly developed countries, absolute decoupling is frequently observed, although only when the DMC indicator is used.

**Keywords:** fossil fuel use; Tapio decoupling analysis; DMC—domestic material consumption; MF—material footprint

## 1. Introduction

Nowadays green growth or "green economy" is one of the most important global concerns. The concept of green growth is currently being promoted by many international institutions and organisations (e.g., the OECD, the United Nations Environment Program—UNEP, and the World Bank) and is also being implemented in various national and international policies (e.g., in the European Union). It is based on the assumption that absolute decoupling of economic growth from resource use or carbon emissions is possible and can take place at a rate that will prevent further climate change and other types of ecological disasters [1]. Comparing the pace of economic growth and the resource use growth indicates different decoupling states. The most desirable one is observed when economic growth is accompanied by a decrease in the use of resources, and is called absolute decoupling. A less favourable situation is observed when the speed of economic growth is positive and faster than the increase in resource use, which is called relative decoupling.

A set of various indicators is proposed to measure green growth (see: [2]). Some green growth indicators are also considered in the literature, but the authors do not agree on how to effectively measure resource use in economy. Two main approaches to measuring it are production-based (territory-based) accounting and consumption-based accounting [3,4]. Most studies devoted to green growth assume that domestic material consumption is the best measure of the total amount of materials used in the economy. This measure was developed within the so-called economy-wide material flow accounts (EW-MFA) [5–7]. Domestic material consumption, or total weight of raw materials

(biomass, minerals, metals, and fossil fuels), is defined as raw materials extracted from the domestic territory plus all physical imports and minus all physical exports. However, the domestic material consumption indicator does not include extractive raw materials related to imports and exports from outside the local economy [8–10]. Therefore, Wiedmann et al. [8] suggest using material footprint as a consumption-based indicator of resource use, which allows taking into account the total impact of consumption on the resources of a given country. It measures the link between the beginning of a production chain (in which raw materials are extracted from the natural environment) and its end (in which a product or service is consumed).

Between 1992 and 2017, global resource use grew by 108%: from 44.2 billion tonnes in 1992 to 91.9 billion tonnes in 2017, and global fossil fuel use increased by 67%: from 8.9 billion tonnes (i.e., 20% of global resources use) in 1992 to 14.9 billion tonnes (16%) in 2017. However, an important distinction should be made when considering the way of measuring fossil fuel use. For example, in this period in high income countries, the domestic material consumption (DMC) of fossil fuels increased by 28.6%: from 4.2 billion tonnes to 5.4 billion tonnes. On the other hand, the material footprint (MF) of fossil fuels increased by 56.3%: from 4.8 billion tonnes to 7.5 billion tonnes. Moreover, in 1992, the MF was 13.4% higher than the DMC, while in 2017, MF was as high as 37.8% of the DMC.

In the contemporary world, renewable energy source use is intensively promoted, which is linked to the decarbonisation process and the need to mitigate carbon emissions. One of the targets of green economy is to reduce the negative environmental impact generated by the use of natural resources in developing economies. This is associated with a reduction in energy consumption, and, more precisely, with an appropriate use of energy resources. That is why it seems justified to analyse the changes in the use of fossil fuels over the last 25 years in countries with different levels of development. The question that needs answering is whether rich countries with a high level of development have actually limited their fossil fuel use or whether they have outsourced a significant part of their production to poorer countries, as a result of which fossil fuel use has moved beyond their balance sheet. Therefore, the examination of decoupling is a key task in the transition to a more resource-efficient economy. This task is also related to the United Nations Sustainable Development Goals, especially clean energy (SDG 7), as fossil fuels still are the main source of energy worldwide.

The abundant literature which studies decoupling in different countries can be divided into two main strands.

The more frequent strand concerns decoupling economic growth from carbon dioxide emissions. De Freitas and Kaneko [11] provide a literature review of the works published by 2010, and Wang et al. [12], Leal et al. [13], and Vadén et al. [14] offer an updated literature review.

Decoupling economic growth from resource use is analysed within the other strand. Wiedenhofer et al. [15], Haberl et al. [4], and Vadén et al. [14] provide an extensive literature review on this subject.

Studies on decoupling economic growth from resource use are conducted at various levels: global [8,16–19], regional (e.g., European Union countries [20,21], OECD countries [12], or BRICS countries [12], 39 countries [22]), or national [23–27]. Some attention is devoted to fossil fuel-exporting countries [28], or consider the resources curse in this context [29].

In most studies, the domestic material consumption indicator is used as a measure of the materials used ([12,19–23]). However, only some studies take into account outsourcing of the production and trade in raw materials (i.e., the material footprint) in measuring the amount of materials used in the economy ([7,8,12,18,30,31]).

In order to investigate the process of decoupling, a considerable number of studies apply descriptive trend analyses ([8,18,32]), while others use econometric techniques (e.g., panel data analysis [21,22]) or other statistical methods (decoupling indicator derived from the IPAT (impact on the environment (I) = population (P), affluence (A) and technology (T)) equation [12,23]).

Some authors analyse whether it is possible to decouple economic growth from resource use [1,33]. Generally, most studies indicate relative decoupling [7,18,26], but, for example, Ward et al. [32] emphasise

the fundamental impossibility of decoupling on an aggregate global scale. Steinberger et al. [22] find the absolute long-term decoupling of domestic material consumption in Germany, the UK, and the Netherlands. Wang et al. [23] observe greater decoupling of resource use from GDP in two OECD countries than in two BRIC countries. Wang et al. [12] report that it is more beneficial for developing countries to decouple economic growth from material footprint than from domestic material consumption, while the opposite is better for developed countries. Similar results are obtained by [8].

It follows from the above considerations that most studies focus on the analysis of decoupling GDP growth from resource use. Only a few studies [12,22,34] examine the use of raw materials while taking into account various categories of these raw materials, such as biomass, non-metallic minerals, fossil fuels, and metal ores. However, to the best of our knowledge, there are only several studies that investigate the problem of decoupling economic growth from fossil fuels on a global scale.

This study aims to fill this gap by conducting in-depth analysis of the decoupling of economic growth from fossil fuel use in 141 countries characterised by different levels of development over the last 25 years (1992–2017). The study is divided into four main stages (see Figure 1). The first focuses on analysing decoupling economic growth from fossil fuel use in countries with different levels of development over the last 25 years. The Tapio decoupling analysis method [35] is used to compare the processes of decoupling economic growth from fossil fuel use in each country. Both approaches, i.e., production-based and consumption-based accounting, are applied to measure the total amount of fossil fuels used. The evaluation of the decoupling processes is performed in ten-year rolling subperiods. Then, the transitions from one state to another are assessed by a first-order Markov chain model. The objective of the next stage is to clarify whether decoupling GDP from fossil fuel use depends on income and/or human development. Time series of decoupling trends for each country are clustered. The k-medoids method (the partitioning around medoids—PAM) [36] is applied because it assigns each object to exactly one cluster. Each cluster is represented by its most centrally located object, which is crucial for interpretation. The resulting groups of countries with similar decoupling are then compared with groups of countries with similar income or with groups of countries with similar human development.

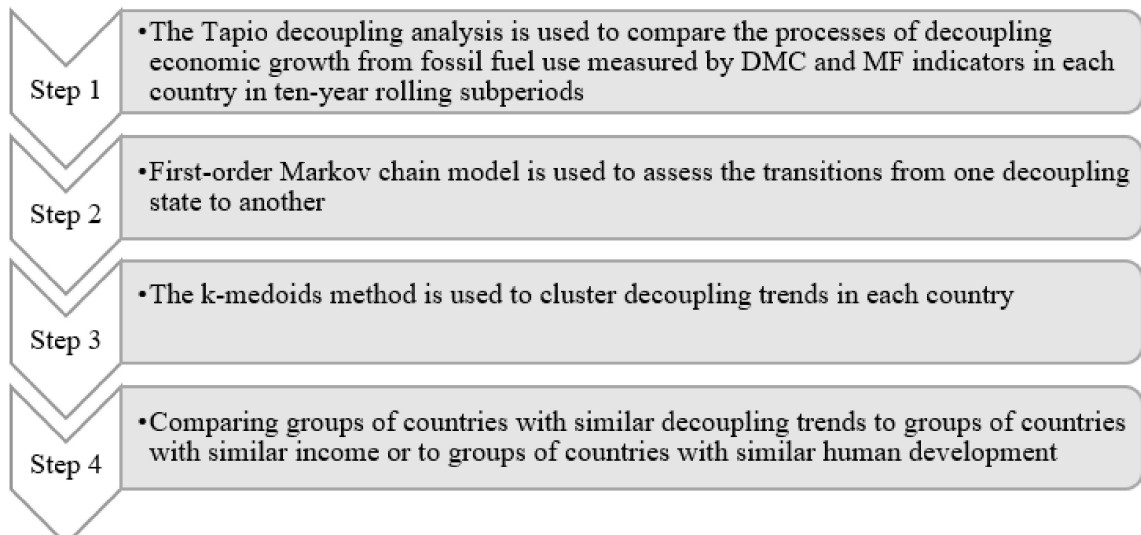

**Figure 1.** Flow chart of global research.

Our results demonstrate significant differences between the decoupling states depending on whether fossil fuel use is measured with domestic material consumption (DMC) or with the material footprint (MF). First, the distributions of the decoupling states in analysed countries differ depending on which measurement method is used. Second, in the case of MF measurement, countries change their decoupling states more often. Third, the relationships between the decoupling patterns in particular

countries and their level of development are also linked to the way in which fossil fuels are measured. Finally, absolute decoupling identified in the recent years by DMC measurement is not confirmed by MF measurement, particularly in a group of high-income countries.

The novel aspects of this study can be summarised as follows.

The first novel aspect is linked to the comprehensive data the study provides regarding the progress in decoupling economic growth from fossil fuel use in countries with different levels of development over the last 25 years, including the countries' transition from one state to another.

Another novelty is the decision to conduct the study in ten-year rolling windows. The rolling windows allow for ignoring the impact of economic fluctuations and climatic factors that could distort the results in short subperiods. Following Tapio [35], the longer the observation period (preferably 5–10 years), the more stable the decoupling trend. Moreover, the ten-year horizon is long enough to observe the effects of the implementation of sustainable development policies, which is a long-term process.

Finally, the novelty of this study lies in the application of two approaches to measure fossil fuel use, which allows the comparison of the decoupling process obtained for the consumption and production approaches. Thus, it is possible to assess whether the reduction in fossil fuel use is a consequence of global green growth, especially in high-income countries or in countries with a very high human development index, or whether absolute decoupling in these countries may simply result from the fact that developed economies "export" their dirty industry to developing economies [37]. When both measurements are used, the results are resistant to a potential impact of outsourcing. Thus, decoupling in the most developed countries measured by MF should be weaker than this measured by DMC.

The remainder of the study is organised as follows. Section 2 presents the methodology and describes the data, Section 3 reports the empirical results, and the last section contains comments and conclusions.

## 2. Materials and Methods

### 2.1. Tapio Decoupling Indicator

An unambiguous definition of decoupling does not exist. Definitions proposed by various organisations, e.g., OECD [38] or the United Nations Environment Programme (UNEP) [39], differ from those used in the literature. Tapio [35] proposes a decoupling indicator that divides the decoupling state into eight types. Lu et al. [40] and Wang et al. [23] define a decoupling indicator based on three grades of decoupling. The Tapio decoupling indicators are used in this study to examine the decoupling of economic growth (G) from resource use (R, domestic material consumption—$R_{DMC}$, material footprint—$R_{MF}$). To calculate the decoupling indicator $\beta_R$, the following formula is used:

$$\beta_R \;=\; \frac{\Delta R / R^B}{\Delta G / G^B} \;=\; \frac{\left(R^t - R^B\right)/R^B}{\left(G^t - G^B\right)/G^B} \tag{1}$$

where $\Delta R$ denotes the changes in resource use, $\Delta G$ is the difference between gross domestic product (GDP) in the audited year in relation to the base year, $B$ is the initial year, and $t$ is the end year.

The decoupling states represented by $\beta_R$ are presented in Table 1; they are classified into three categories and eight sub-categories.

Numerous authors modify Tapio's classification in different ways, for example, Naqvi and Zwickl [41], whose proposal is followed in this study, use five states. The most advantageous situation is absolute decoupling (AD), in which the growth of GDP is accompanied by a decrease in resource use. Relative decoupling (RD) means that an increase in resource use is accompanied by faster GDP growth, while coupling (C) is a situation in which GDP growth is accompanied by faster growth of resource use. The most unfavourable situation, called negative decoupling (ND), happens when, despite the increase in resource use, GDP decreases. From the point of view of economic development, negative coupling (NC)—when both resource use and GDP decrease—is also unfavourable.

**Table 1.** Tapio's classification of decoupling states.

| State | | $\Delta R/R^B$ | $\Delta G/G^B$ | $\beta_R$ |
|---|---|---|---|---|
| Negative decoupling | Expansive negative decoupling | + | + | $(1.2, +\infty)$ |
| | Strong negative decoupling | + | − | $(-\infty, 0)$ |
| | Weak negative decoupling | − | − | $[0, 0.8)$ |
| Decoupling | Weak decoupling | + | + | $[0, 0.8)$ |
| | Strong decoupling | − | + | $(-\infty, 0)$ |
| | Recessive decoupling | − | − | $(1.2, +\infty)$ |
| Coupling | Expansive coupling | + | + | $[0.8, 1.2]$ |
| | Recessive coupling | − | − | $[0.8, 1.2]$ |

### 2.2. Markov Chain Model

The first-order Markov chain model is used to assess the transition from one state to another. It is assumed that a country might move across five states S = {AD, RD, C, ND, NC} and the probability of transition from an initial state to other state $(i, j)$ is time invariant:

$$P(X_1 = j | X_0 = i) = p_{ij} \tag{2}$$

We focus on the one-step transition probability matrix:

$$P = \begin{bmatrix} p_{11} & p_{12} & p_{13} & p_{14} & p_{15} \\ p_{21} & p_{22} & p_{23} & p_{24} & p_{25} \\ p_{31} & p_{32} & p_{33} & p_{34} & p_{35} \\ p_{41} & p_{42} & p_{43} & p_{44} & p_{45} \\ p_{51} & p_{52} & p_{53} & p_{54} & p_{55} \end{bmatrix} \tag{3}$$

in which rows sum up to unity. Two P matrices are considered: one for domestic material consumption $R_{DMC}$ and one for material footprint—$R_{MF}$.

### 2.3. The PAM Clustering Method

Clustering techniques are applied to find distinct groups of countries which share similar trends of decoupling states. Each country is represented by the vector of decoupling states in the period 2002–2017. As the vectors consist of nominal variables, all five categories are first converted into five binary columns. Next, the Dice coefficient, which is equivalent to the Gower distance [42] for categorical variables, is used to calculate the distance between the countries. Finally, based on the distance matrix, the partitioning among medoids (PAM) algorithm [36] is used to find the clusters. The PAM algorithm is similar to the *k*-means method, and they share the same aim: to cluster the *n*-object into *k* clusters (the assumed number of *k* is known a priori). In contrast to the k-means algorithm, PAM treats the elements of the sample as centres (medoids) of clusters. These medoids can be identified, as opposed to the k-means, as elements from the sample. This way, the decoupling trends of the most central countries in every group can be analysed. The average silhouette method [43] is used to decide on the number of clusters (*k*).

### 2.4. Data

The analysis is conducted in 141 countries in the period between 1992 and 2017. Three yearly series are collected: the first two are fossil fuel use measured twice, and the third one is economic growth. The time framework used in the study is dictated by the data availability, and, following [35], the decoupling states are calculated in ten-year rolling windows. The first decoupling state is obtained for the period 1992–2002, the second for the period 1993–2003, etc. Finally, each country is described by the vector that consists of sixteen elements, and each element represents a decoupling state in

a subsequent window. This vector reveals the decoupling patterns that a given country followed. Moreover, in order to observe the changes, the entire period is divided into ten-year rolling windows.

Gross domestic product (GDP) in constant prices (constant 2015 US dollar) is used to measure economic output. Two kinds of measurements of resource use are applied, and only fossil fuels are taken into account in the analysis. The assessment of resource consumption takes into account both sets of data: those related to domestic material consumption of fossil fuels (DMC is based on official economic statistics and requires some modelling to adapt the source data to the methodological requirements of the MFA.) and those related to material footprint of fossil fuels (MF), i.e., the allocation of global material extraction to the domestic final demand of a country. The total material footprint is the sum of the material footprint for biomass, fossil fuels, metal ores, and non-metal ores. It is calculated as the raw material equivalent of imports (RMEIM) plus domestic extraction (DE) minus raw material equivalents of exports (RMEEX). A global, multi-regional input–output (MRIO) framework is employed to attribute the primary material needs of final demand. The attribution method based on I–O analytical tools is described in detail in Wiedmann et al. [8]. It is based on the EORA MRIO framework developed by the University of Sydney, Australia [44], which is a global, well-established, and the most detailed and reliable MRIO framework available to date.

The datasets are obtained from the United Nations Environment Programme (UNEP) database. UNEP has the specific mandate to develop SDG methodologies, training materials for countries, and SDG data reporting mechanisms.

In order to understand the decoupling patterns of each country, their level of development is analysed with the use of two indicators. One is income (INC): it measures gross national income (GNI) per capita, in US dollars, converted from local currency using the World Bank Atlas method. The other, proposed by [45], is the human development index (HDI): it measures average achievements in key dimensions of human development, i.e., a long and healthy life, being knowledgeable, and having a decent standard of living. The data come from the World Bank's World Development Indicators (WDI) database [46] and the United Nations Development Programme (UNDP) database [47], respectively.

Next, the countries are classified according to their income (INC) and the HDI. Following the classifications of the World Bank, four income groups are considered: low (L_INC), lower-middle (LM_INC), upper-middle (UM_INC), and high (H_INC) income, and—following the classifications of UNDP—four categories of human development are used: low (L_HDI), medium (M_HDI), high (H_HDI), and very high (VH_HDI) human development. The detailed description of cut-off points of the HDI for grouping countries is given in the Human Development Report [48].

Figure 2 presents the average per capita fossil fuel use in the following years for the countries classified in the individual groups from the point of view of income and the HDI.

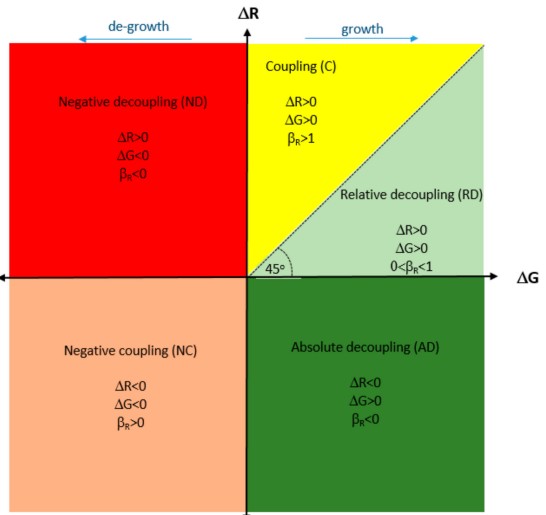

**Figure 2.** Five decoupling states in Tapio's classification.

Figure 3 clearly demonstrates that there are significant differences in average fossil fuel use depending on the countries' level of development. In the case of both income and the HDI, highly developed countries on average use significantly more fossil fuels than less developed countries.

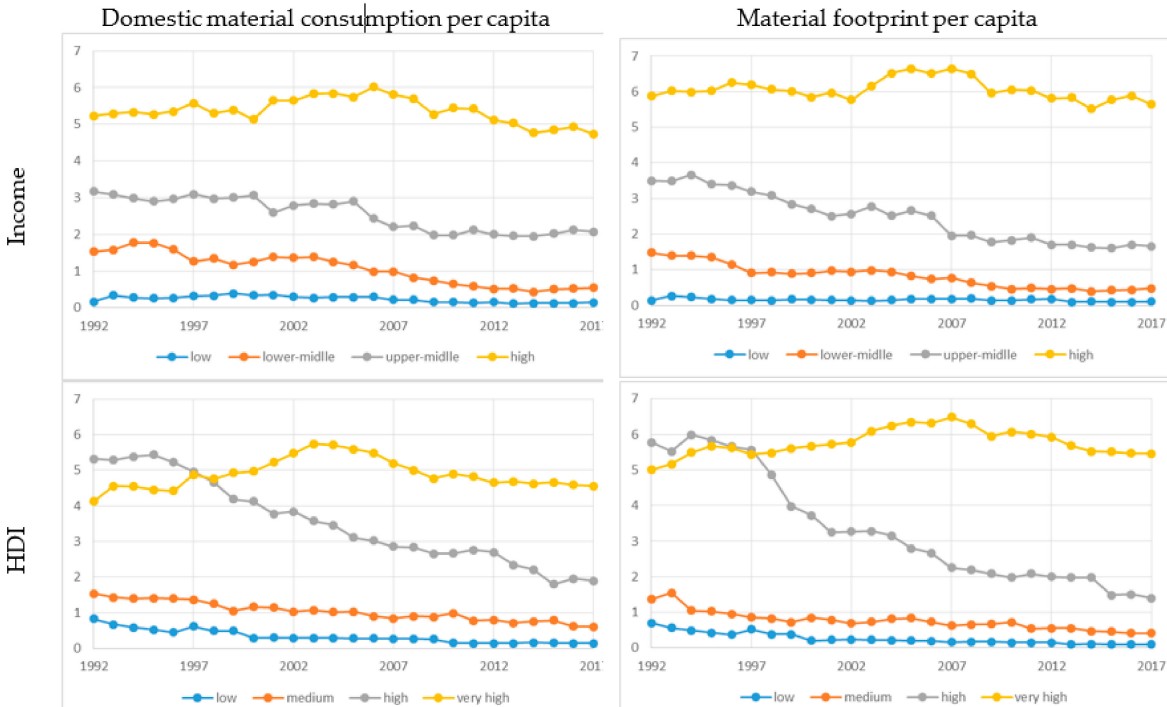

**Figure 3.** Per capita domestic material consumption and material footprint (fossil fuels) by income and human development index (HDI) level, 1992–2017.

In the entire period of analysis, the highest average fossil fuel use measured both by domestic material consumption per capita (DMCpc) and material footprint per capita (MFpc) is observed in the high-income countries, although since 2006, the average value of fossil fuel use in these countries has been decreasing. Both measurements also show a negative trend in the average fossil fuel use in the upper-middle income countries. Decisively lower average fossil fuel use is noted in less developed countries (those with low and lower-middle income). Moreover, in the recent period, a slight increase in the average resource use is observed in these groups.

Similar conclusions can be drawn when the HDI classification is adopted: the highest average value of resource use is found in the most developed countries (those with very high human development). Both measurement methods reveal a distinct negative trend in the average values in countries characterised by high human development. Before 1998, the average use of fossil fuels in these countries is even higher than in the group of highly developed countries. Over time, however, it decreases to a level slightly higher than in the least developed countries.

It should also be stressed that in high-income countries (and in those with very high human development), the fossil fuel use measured by MFpc is slightly higher than the one measured by DMCpc, which may confirm the thesis that highly developed countries use outsourcing.

## 3. Results

### 3.1. Tapio Results

The time series of individual decoupling trends obtained by applying the five-level Tapio decoupling model are presented in Figure 4. The left panel presents the results linked with domestic material consumption, while the right is related to material footprint. Each cell in the diagram shows the

decoupling state calculated for a ten-year period. For example, the first cell that refers to Afghanistan in the left panel (C) indicates the decoupling of economic growth from DMC in the period 1992–2002. The whole vector shows the decoupling trend this country followed in the period 1992–2017.

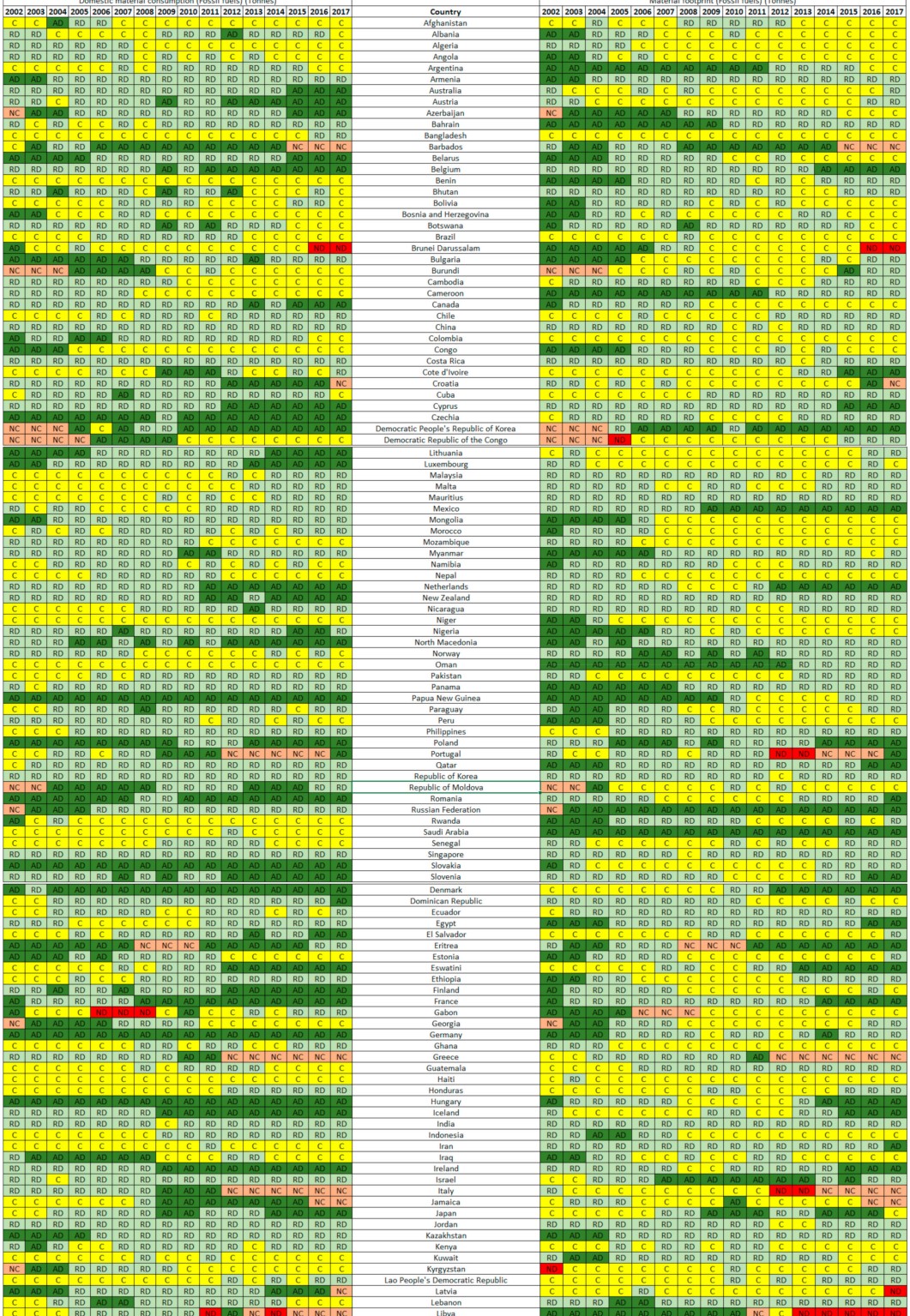

**Figure 4.** *Cont.*

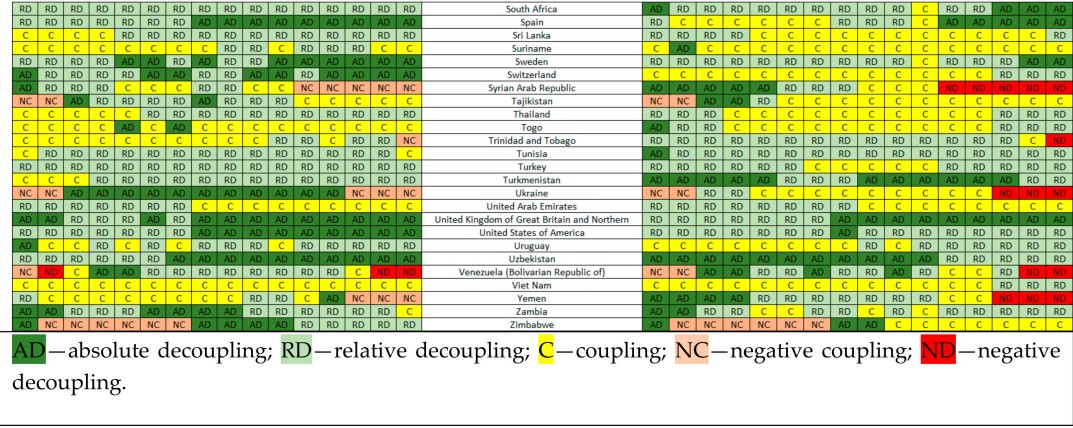

AD—absolute decoupling; RD—relative decoupling; C—coupling; NC—negative coupling; ND—negative decoupling.

**Figure 4.** Decoupling trends in ten-year rolling windows from 1992–2017 and two methods of measuring fossil fuel use.

The most important observation derived from the analysis of Figure 4 is that in most cases, the growth of fossil fuel use is accompanied by GDP growth (C and RD states). In some countries, the situation is favourable because GDP growth is faster than the growth of resource use (RD is 1033 (46%) cases for DMC and 979 (43%) cases for MF), but in others, GDP growth is associated with a greater increase in fossil fuel use (C is 652 (29%) cases for DMC and 817 (36%) cases for MF). The situation of the most favourable AD cases for MF is more common when fossil fuel use is measured using DMC (AD is 491 (22%) cases for DMC and 380 (17%) cases for MF). However, there are few countries in which absolute decoupling is observed over the entire period (Germany, Hungary, Papua New Guinea, Saudi Arabia), and AD appears in them more frequently in the last ten-year periods examined. The most unfavourable ND occurs sporadically in individual countries in single ten-year periods: 10 (0.4%) cases for DMC and 28 (1.2%) cases for MF. MF is also more frequent in the last years analysed in the study.

The share of individual decoupling states in ten-year rolling windows is presented in Figure 5. Figure 5a demonstrates the shares of the individual decoupling states in all countries; next, the countries are divided by the income criterion, and Figure 5b shows these shares in high-income (H_INC) countries, while Figure 5c shows them in low- (L_INC), lower-middle- (LM_INC), and upper-middle-income (UM_INC) countries.

(**a**) in all countries

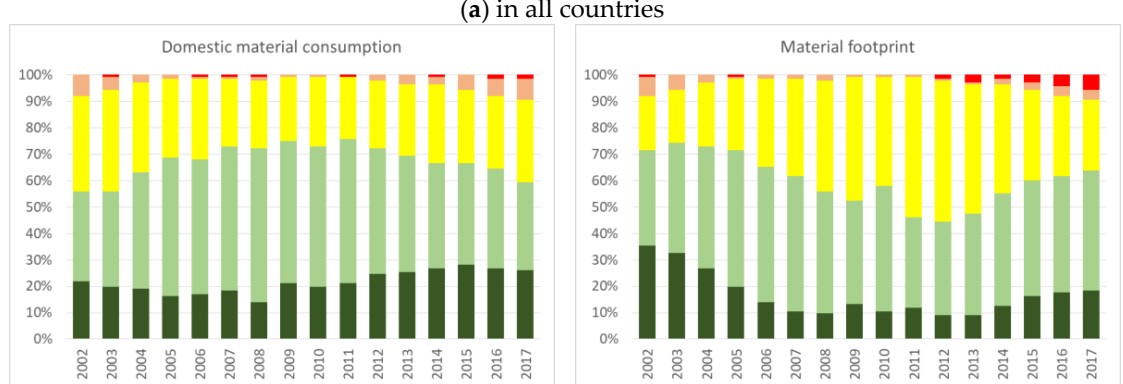

**Figure 5.** *Cont.*

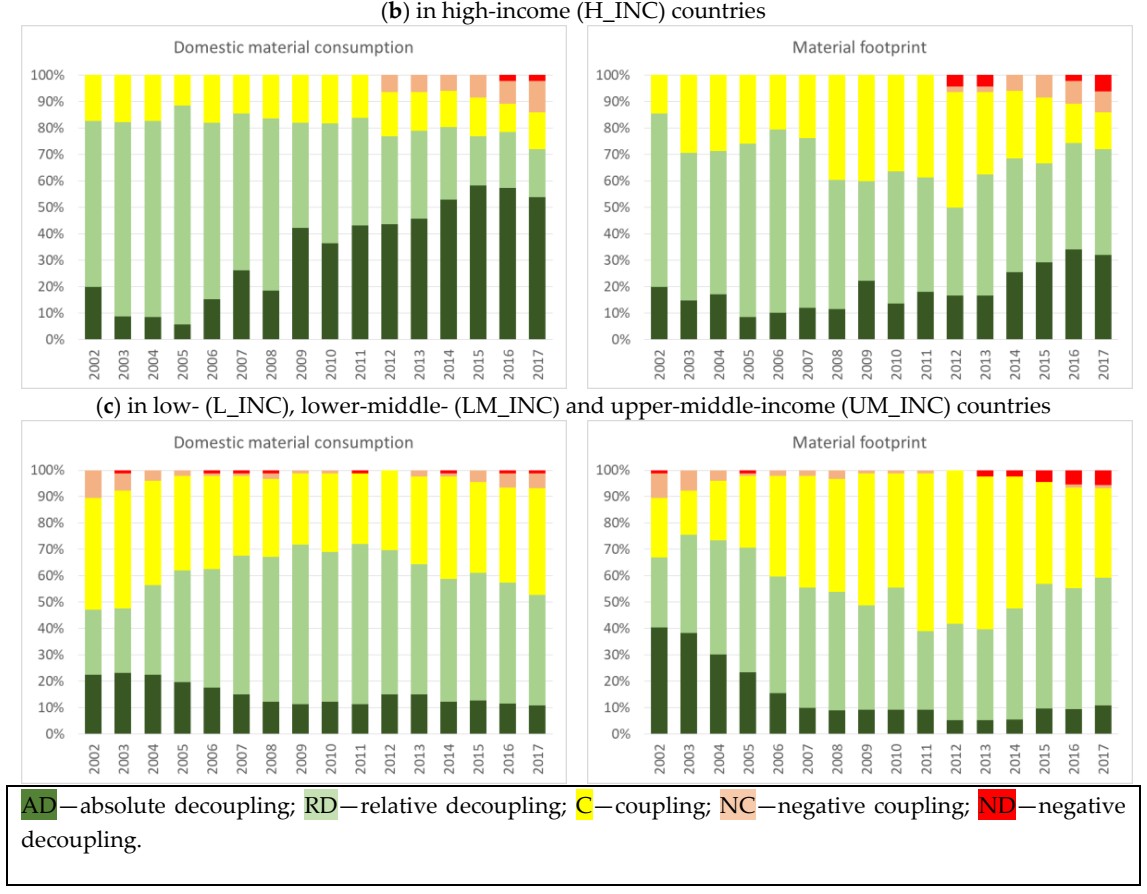

**Figure 5.** The share of decoupling states in ten-year rolling windows from 1992–2017 obtained by two methods of measuring fossil fuel use.

Figure 5 clearly demonstrates the prevalence of the simultaneous growth of fossil fuel use and GDP in all ten-year rolling windows. For DMC, the share of countries with more favourable RD prevails, with more than 50% in some ten-year rolling windows. In the case of MF, there are ten-year rolling windows in which C is dominant, i.e., the growth of fossil fuel use is faster than the growth of GDP.

In the case of DMC, the share of countries (Figure 5a and Table A1 in Appendix A) in which C is observed is almost constant in subsequent ten-year rolling windows (the smallest share—23% of all cases—is found in the 2001–2011 period and the largest share—38%—in the 1993–2003 period). In the case of MF (Table A2 in Appendix A), this share increases significantly in the ten-year rolling windows which end between 2010 and 2012 (from 41% to 53%) and, although it subsequently decreases slightly, it is still higher than in the initial period of the study (21% of all cases in the first period and 27% in the last period). A slow but considerable increase in the number of countries with AD is observed. In the case of DMC, the share of countries with AD is at a similar level: about 20% in initial ten-year rolling windows and then 26%. In the case of MF, the share of countries with AD is about 30% in the initial ten-year rolling windows; next, this share drops to 10% in ten-year rolling windows that end between 2007 and 2013. In the last period, their percentage increases to 20%, although it is still lower than at the beginning of the study.

In the case of DMC, countries with high income have a large share of AD, especially in recent years (see Figure 5b). In the last ten-year rolling window, this share equals 54% and increases mainly as a result of the decline in the share of RD, which means that fossil fuel consumption in most countries decreases. Recently, most countries with AD belong to the EU: Austria, Belgium, Cyprus, Czech Republic, Denmark, Finland, France, Germany, Hungary, Ireland, Lithuania, Luxembourg, Netherlands,

Romania, Slovakia, Slovenia, Spain, Sweden, and the UK, but also Australia, Azerbaijan, Belarus, Canada, Korea, Dominican Republic, El Salvador, Eswatini, Iceland, Japan, New Zealand, Switzerland, USA, and Uzbekistan. The share of countries with C is at a similar level (about 15%), and they include: Argentina, Estonia, Kuwait, Norway, Oman, Saudi Arabia, and the United Arab Emirates. In the case of MF, the share of countries with AD is much lower—32%, and they are characterised by faster GDP growth and slower growth of fossil fuel use (RD is 40%). The countries with AD in recent years include: Belgium, Cyprus, Denmark, France, Hungary, Ireland, the Netherlands, Poland, Slovenia, Spain, Sweden and the UK, so mainly the EU member states, and, in addition: Iceland, Portugal, Qatar, and Saudi Arabia. The share of countries with C is similar to that of DMC.

When DMC is taken into account in the group of low-, lower-middle-, and upper-middle-income countries (see Figure 5c), their biggest share is RD (recently over 40%) and C (also about 40%). AD is found in much fewer countries: Azerbaijan, Belarus, Dominican Republic, El Salvador, Eswatini, North Macedonia, Papua New Guinea, Romania, and Uzbekistan. In the case of MF in the middle ten-year rolling windows, C (about 60%) is observed in most countries, although it again changes to RD (almost 50%) in recent years.

There are only a few countries with ND, but their percentage is increasing, especially when MF is the basis for measuring fossil fuel use (from 1% in the beginning to 6% in the last period). In addition, it should be noted that negative decoupling in the last period is observed mostly in high-income countries. The percentage of these two states in the last period is 14% (regardless of the measurement method used) in the group of H_INC countries, while in the group of low-, lower-middle-, and upper-middle-income countries, it is only 6%.

### 3.2. Markov Chain Results

The decoupling states in ten-year periods indicated in Figure 4 do not show how stable these states are in particular countries. In particular, it is not known whether a given state of decoupling in subsequent periods has been identified for the same or for other countries. In order to check this, a first-order Markov chain model is used in the next step to assess the probability of transition from one state to another.

Figure 6 presents the one-step transition probability calculated for five decoupling states for DMC. The arrows and the numbers indicate the probability of a move from one state to another. Several conclusions can be made here. First, the states are relatively stable. The probability that a country will remain in a given state for the next period is always higher than the probability that it will leave it. To most stable state is RD (0.82), which means that 82% of countries that are classified as RD in a given period remain in RD for the next period. The probability of remaining in AD and C equals 0.8. NC, which appears to be the least stable state, has lower probability—0.5, which means that countries can easily leave this state.

The following observations can be made when transitions between different states are considered. Countries classified as ND in the initial period have the largest chance (0.3) to move to the C state. If countries are to leave the C state, they will most probably move to RD, and if they are to leave the RD state, they will move either to C (0.10) or AD (0.08). If countries leave AD, they usually (0.14) move to RD, while countries classified in the initial period as NC frequently move to AD (0.21).

Figure 7 presents the one-step transition probability calculated for five decoupling states for MF. In comparison to the transitions obtained for DMC, some differences can be noticed. First, the chances to remain in the same state are smaller than in the case of DMC. These chances are 0.79 for RD, 0.78 for C, 0.73 for AD, 0.64 for NC, and 0.5 for ND. Consequently, transitions between states are more frequent. If countries are classified as ND in the initial state, they have a great chance (0.29) of moving to NC in the next period. Countries in the NC state usually (0.21) move to AD, and from AD they most frequently (0.22) go to RD. If countries change their state beginning from RD in the initial period, most of them go to C (0.14). Interestingly, the move from C to RD is also quite frequent (0.16).

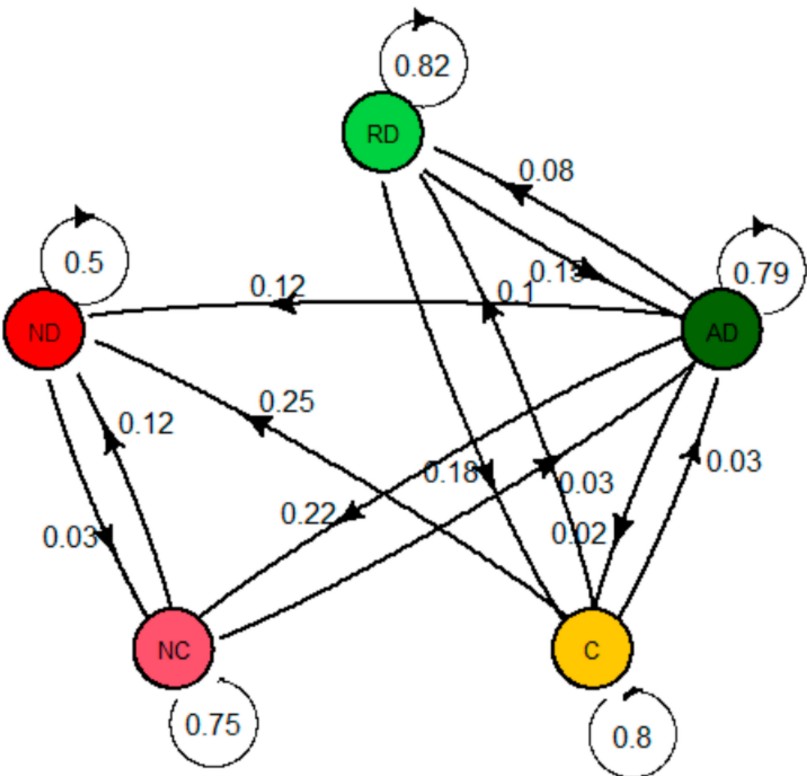

**Figure 6.** One-step transition probability plot estimated for domestic material consumption (DMC) decoupling trends.

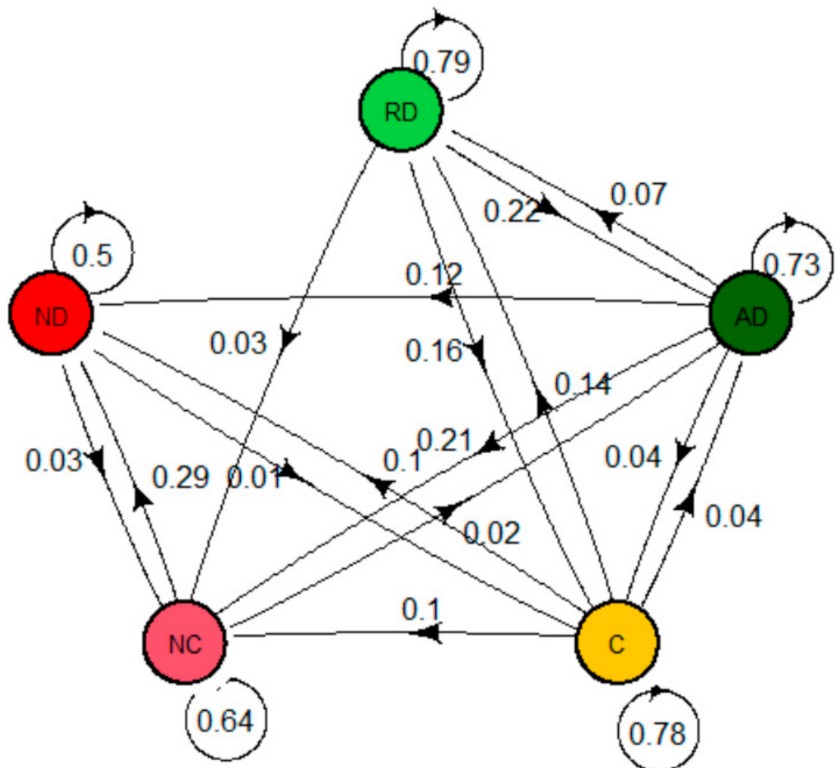

**Figure 7.** One-step transition probability plot estimated for material footprint (MF) decoupling trends.

### 3.3. Clustering Results

The PAM algorithm is used to cluster decoupling trends of the analysed countries. It is assumed that each country is represented by the vector of decoupling states, which is presented in Figure 4. When comparing the grouping results by the silhouette average width, between two and six clusters are considered. The silhouette criterion yields a similar division, although the optimal number of clusters in the case of DMC is three, while in the case of MF—four. The results obtained in these optimal divisions are presented below.

Table 2 summarises the clustering results obtained for the DMC. It shows the number of countries in each cluster and their medoids (the most central element in each group). The second group is the largest one, as it includes 69 countries, while the remaining groups contain 39 countries. Table 2 also presents the decoupling trend for all the medoids. In Vietnam (the medoid of the first cluster), coupling (C) between GDP growth and fossil fuel use is observed in every window. In Turkey, coupling (C) in the initial two subperiods and relative decoupling in the remaining subperiods is noted. In the US, relative decoupling is found in the first nine subperiods and absolute decoupling in the next nine subperiods. Countries assigned to different development levels are the central elements of each cluster. According to the World Bank classification (we apply the classification from 2017), Vietnam is a low-income country, Turkey is an upper medium-income country, and the US is a high-income country. The chi-square test is used to find the relationship between the trend of decoupling, which is represented by the group of countries obtained by means of PAM, and the development level of these countries. The obtained result ($p$-value 0.00000003) indicates a strong relationship between them. Figure 8 depicts the relationship between these two variables. Most elements of the first group, with Vietnam as the central element, are classified as low-income countries. The elements of the second group (with Turkey as the medoid) are usually classified as LM_INC or UM_INC, while countries from the third group are frequently classified as high-income ones.

**Table 2.** The number of countries in each group, the medoids, and the decoupling trend of the medoids obtained for DMC.

| Group | No of Countries | Medoid | Decoupling Trend |
|---|---|---|---|
| 1 | 39 | Vietnam | C C C C C C C C C C C C C C C C |
| 2 | 63 | Turkey | RD RD RD RD RD RD RD RD RD RD RD RD RD RD RD RD |
| 3 | 39 | United States of America | RD RD RD RD RD RD RD AD AD AD AD AD AD AD AD AD |

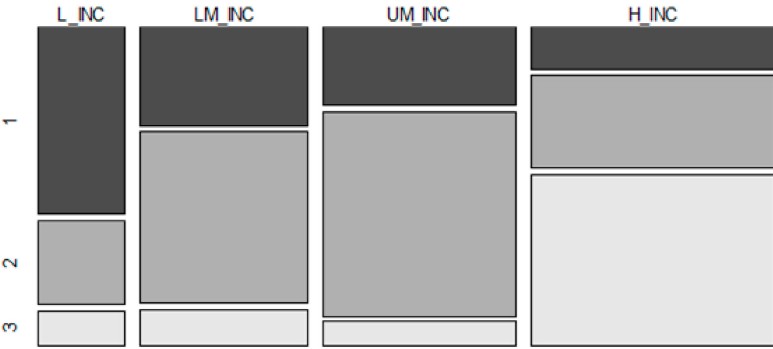

**Figure 8.** Relationship between groups of countries with a similar decoupling trend obtained for DMC and income groups. Note: The chi-square test indicates the rejection of independence of the two variables (chi-square = 45.663, $p$-value = $3.455 \times 10^8$).

The relationship between groups of countries with a similar decoupling trend obtained for DMC and the HDI are also tested. The mosaic presented in Figure 9 visualises this relationship. The results are similar to those presented in Figure 8. The countries from the first group are most frequently classified as low HDI (L_HDI). The higher the countries' development level, measured by the HDI, the fewer countries belong to the first group. Most countries from the second group belong to the M_HDI or H_HDI groups. Finally, the countries from the last group are usually VH_HDI. The chi-square test indicates that the relationship between the HDI and group categories is highly significant (*p*-value = 0.0000003).

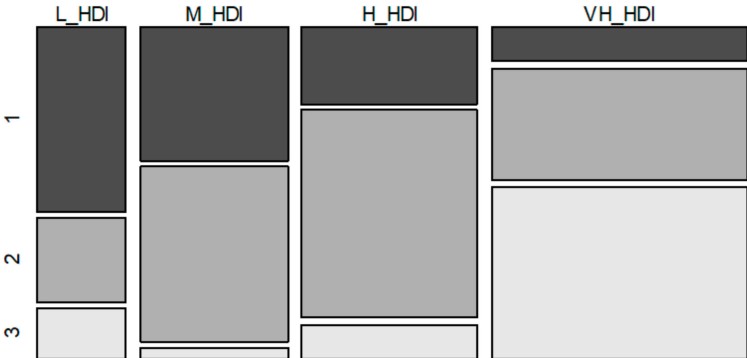

**Figure 9.** Relationship between groups of countries with a similar decoupling trend obtained for DMC and the HDI. Note: The chi-square test indicates the rejection of independence of the two variables (chi-square = 46.333, *p*-value = $2.541 \times 10^8$).

Next, the results obtained for the decoupling of economic growth from fossil fuel use measured as MF are presented. The strategy applied is the same as in the case of fossil fuel use measured as DMC. First, the clustering is performed in order to obtain the groups of countries with a similar decoupling trend, and next these groups are compared with the classification of income levels and country development levels.

Table 3 presents the number of elements in groups and the medoid of each group. Group four and group one are the most numerous: the former (with India as the medoid) consists of 55 countries, and the latter (with Lithuania as the medoid)—of 44 countries. Group three (with Oman as the central element) consists of 14 countries and is the least numerous. Group two includes 28 countries. Table 3 also shows the trends in the decoupling states of medoids. In the case of the medoid of the first group (Lithuania), coupling is observed in most periods. The central element of the second group (Peru) begins with absolute decoupling (AD) and relative decoupling (RD) and ends with coupling (C). The medoid of the third group (Oman) is classified in most periods as absolute decoupling but ends with relative decoupling. Finally, the medoid of the fourth group (India) is classified as relative decoupling in all periods.

**Table 3.** The number of countries in each group, the medoids, and the decoupling trend of the medoids obtained for MF.

| Group | No of Countries | Medoid | Decoupling Trend |
|-------|-----------------|--------|------------------|
| 1 | 44 | Lithuania | C RD C C C C C C C C C C C C RD RD |
| 2 | 28 | Peru | AD AD AD RD RD RD RD C C C C C C C C C |
| 3 | 14 | Oman | AD AD AD AD AD AD AD AD AD AD AD RD RD RD RD RD |
| 4 | 55 | India | RD RD RD RD RD RD RD RD RD RD RD RD RD RD RD RD |

Several important remarks should be made here. The medoids of the groups no longer belong to the different income groups. Lithuania and Oman are classified as high-income countries, Peru as an

upper medium-income country, and India as a lower medium-income country. In comparison to the results obtained for DMC, the ones obtained for MF do not contain any group with absolute decoupling in the last subperiods. This means that when material footprint is considered, the GDP growth in the last decade is typically accompanied by the increase in fossil fuel use regardless of the country's income. Second, the changes of the decoupling states from AD to RD and from RD to C are observed, which means that the expectation that over time the economic growth will require increasingly fewer fossil fuels has not been met.

Finally, the relationship between the trends of the decoupling states obtained for MF and two country development classifications are analysed. Figure 10 presents the mosaic that illustrates the decoupling states trend groups versus the income groups.

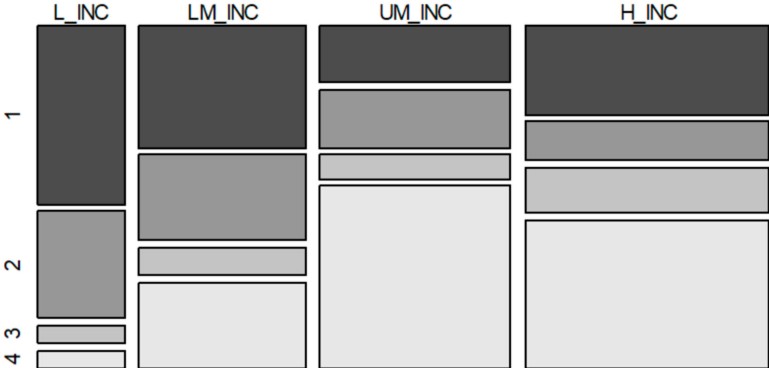

**Figure 10.** Relationship between groups of countries with a similar decoupling trend obtained for MF and income groups. Note: The chi-square test indicates the rejection of independence of the two variables (chi-square = 21.949, *p*-value = 0.009).

In the first group, more countries belong to the low-income and lower medium-income groups. Interestingly, more countries from this group belong to a high-income group than to an upper medium-income group. When countries assigned to the second group are considered, most of them are low- (L_INC) or lower medium-income (LM_INC) countries. Countries from the third group are evenly distributed between income levels, while the fourth group is dominated by upper medium- (UM_INC) and high-income (H_INC) countries. The chi-square test indicates a significant relationship (*p*-value = 0.009) between these two categories.

Figure 11 illustrates the relationship between the trends within the decoupling states in groups and the HDI. To a large extent, the results obtained here are similar to the ones obtained in the income group classification (see Figure 9). This means that countries from the first two groups frequently have low or medium HDI. Still, a large number of countries from these two groups are either in the group with the high HDI (H_HDI) or the very high HDI (VH_HDI). A similar number of countries from the third group belong to the low HDI (L_HDI) and the very high HDI (VH_HDI). Countries from the fourth group are rarely found in the low HDI (L_HDI) group and frequently in the remaining groups. Importantly, the relationship between the HDI categories and group membership is not obvious, as the *p*-value of the chi-square test does not exceed 0.05. This means that, when decoupling is analysed with the use of MF measurement, the countries with the same development levels do not share decoupling trends.

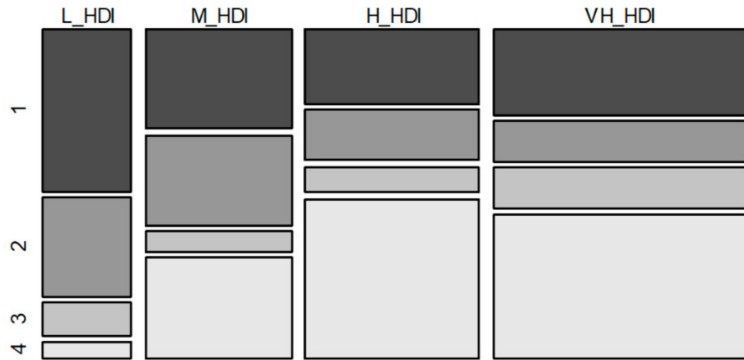

**Figure 11.** Relationship between groups of countries with a similar decoupling trend obtained for MF and the HDI groups. Note: The chi-square test indicates the rejection of independence of the two variables (chi-square = 16.837, *p*-value = 0.051).

## 4. Conclusions and Discussion

This study analyses the process of decoupling economic growth from fossil fuel use in countries with different levels of development over the last 25 years. Two measures of fossil fuel use are applied. The results obtained allow for formulating several conclusions.

The most important one is that the process of decoupling economic growth from fossil fuel use differs when different measures of fossil fuel use are considered. These differences can be spotted in various areas.

Firstly, in general, when the DMC indicator is considered, relative decoupling is the most commonly observed in the analysed countries, while absolute decoupling and coupling appear with similar frequency. When the MF indicator is used, the relative decoupling and coupling happen more frequently. Absolute decoupling of economic growth from fossil fuel use is less frequently observed, especially in recent subperiods. Our results are in line with the results obtained by other authors who examined the existence of decoupling for all types of raw materials. The most frequent relative decoupling obtained in our study, regardless of how fossil fuel use is measured, is consistent with the results reported by [7,18,26].

Secondly, in the case of the MF indicator, a country is less likely to have the same decoupling states in the next subperiod than in the case of the DMC indicator, so the decoupling states of these countries change more frequently. This means that fossil fuel use has greater variability if the entire supply chain is measured. This result is similar to the one obtained by Kan et al. [34], who report that countries frequently change their decoupling states.

Thirdly, the clustering results obtained for decoupling trends also differ. Taking into account DMC, there is one group of countries in which absolute decoupling is observed in the last few subperiods. This group is dominated by highly developed countries, i.e., those with high income or the very high HDI. In this group of countries, economic growth is accompanied by a decrease in fossil fuel use. Taking MF into account, there is no group of countries in which absolute decoupling is observed in the last few subperiods. This means that, regardless of the countries' level of development, economic growth requires an increase in fossil fuel use measured by the entire production chain.

The groups of countries with similar trends in decoupling GDP from fossil fuel use are related to their income level or their human development level. These dependencies are significant when the DMC indicator is taken into account. In this case, coupling is observed in most lower-income countries or those with lower human development, and absolute decoupling is observed in most high-income countries or those with very high human development. Consequently, the expectation that when countries become richer, their fossil fuel use decreases, is generally met. Yet, there are highly developed countries in which absolute decoupling is not observed. When the MF indictor is considered, the relationship between decoupling and income level still exists, although in this case the

relationship pattern is different. In particular, there is no group of countries with absolute decoupling, and highly developed countries usually undergo relative decoupling.

Finally, in the case of the DMC indicator, in high-income countries, absolute decoupling is most frequently observed, particularly in the last subperiods. However, when the MF indicator is considered, highly developed countries are most frequently characterised by the relative decoupling. It is worth noting that in recent subperiods, the share of countries with absolute decoupling increases in the group of high-income countries. When the entire production chain in these countries is taken into account, it turns out that their economic growth is accompanied by lower fossil fuel use. In recent subperiods, high-income countries with such absolute decoupling include Belgium, Denmark, France, Hungary, the Netherlands, Poland, Spain, the United Kingdom, and Saudi Arabia. With the exception of Saudi Arabia, these are all EU countries that implement an energy and climate policy aimed at increasing RES (renewable energy sources) and decreasing $CO_2$. Meeting both these goals has a strong impact on reducing fossil fuel use. However, absolute decoupling is observed in Mexico and Russia, which are lower-income countries. Together with Saudi Arabia, these are major oil-exporting countries (or natural gas or coal—Russia). It is worth considering this factor as the one that determines the state of decoupling in further studies.

To sum up, our results are in line with Wiedmann et al. [8], who found that when the material footprint is used to measure resource use, decoupling between economic achievements and environmental impacts was "smaller than reported or even non-existent", due to the export of production chains to other countries.

Most highly developed counties follow a policy aimed at reducing $CO_2$ emissions and transformation towards a green economy. Indirectly, these policies reduce the use of fossil fuels in the energy or residential sectors. The results of our study reveal some effects of such policies.

The number of rich countries with absolute decoupling obtained using the MF indicator in recent years is significantly lower than those obtained using the DMC indicator. There is, however, a note of optimism in this regard, as the number of countries with absolute decoupling has been increasing steadily with regard to the MF indicator.

We have not been able to fully confirm the results obtained by Wang et al. [12] and Kan et al. [34], according to whom absolute decoupling of resources or energy sources and economic growth should be more frequently observed in developing countries within the MF approach than within the DMC approach. Our results demonstrate that this is the case for fossil fuels but only at the beginning of the study period. Recently, a more or less equal number of lower-income countries have shown signs of absolute decoupling within both approaches.

The question remains why some developed and developing countries reach absolute decoupling when the MF indicator is used. Unfortunately, this question cannot be answered on the basis of the present study. For example, only seven out of twenty-eight EU countries which implement the same climate policy have achieved absolute decoupling. It is possible that all European Union countries are moving towards absolute decoupling, but they need more time to achieve it. The influx of new data and the extension of the study would perhaps help to answer this question.

**Author Contributions:** Conceptualisation, K.F., M.P., and S.Ś.; methodology, K.F., M.P., and S.Ś.; software, K.F., M.P., and S.Ś.; validation, K.F., M.P., and S.Ś.; formal analysis, K.F., M.P., and S.Ś.; investigation, K.F., M.P., and S.Ś.; resources, K.F., M.P., and S.Ś.; data curation, K.F., M.P., and S.Ś.; writing—original draft preparation, K.F., M.P., and S.Ś.; writing—review and editing, K.F., M.P., and S.Ś.; visualisation, K.F., M.P., and S.Ś.; supervision, K.F., M.P., and S.Ś.; project administration, K.F., M.P., and S.Ś.; funding acquisition, K.F., M.P., and S.Ś. All authors have read and agreed to the published version of the manuscript.

**Funding:** This research was funded from the funds granted to the Cracow University of Economics, within the framework of the POTENTIAL Program, project number 38/EIT/2020/POT.

**Conflicts of Interest:** The authors declare no conflict of interest. The funders had no role in the design of the study; in the collection, analyses, or interpretation of data; in the writing of the manuscript, or in the decision to publish the results.

# Appendix A

**Table A1.** The share of the decoupling states in ten-year rolling windows from 1992–2017 for DMC.

| Year | 2002 | 2003 | 2004 | 2005 | 2006 | 2007 | 2008 | 2009 | 2010 | 2011 | 2012 | 2013 | 2014 | 2015 | 2016 | 2017 |
|------|------|------|------|------|------|------|------|------|------|------|------|------|------|------|------|------|
| in all countries | | | | | | | | | | | | | | | | |
| AD | 22% | 20% | 19% | 16% | 17% | 18% | 14% | 21% | 20% | 21% | 25% | 26% | 27% | 28% | 27% | 26% |
| RD | 34% | 36% | 44% | 52% | 51% | 55% | 58% | 54% | 53% | 55% | 48% | 44% | 40% | 38% | 38% | 33% |
| C | 36% | 38% | 34% | 30% | 30% | 26% | 26% | 24% | 26% | 23% | 26% | 27% | 30% | 28% | 28% | 31% |
| NC | 8% | 5% | 3% | 1% | 1% | 1% | 1% | 1% | 1% | 0% | 2% | 4% | 3% | 6% | 6% | 8% |
| ND | 0% | 1% | 0% | 0% | 1% | 1% | 1% | 0% | 0% | 1% | 0% | 0% | 1% | 0% | 1% | 1% |
| in high-income (H_INC) countries | | | | | | | | | | | | | | | | |
| AD | 20% | 9% | 9% | 6% | 15% | 26% | 19% | 42% | 36% | 43% | 44% | 46% | 53% | 58% | 57% | 54% |
| RD | 63% | 74% | 74% | 83% | 67% | 60% | 65% | 40% | 45% | 41% | 33% | 33% | 27% | 19% | 21% | 18% |
| C | 17% | 18% | 17% | 11% | 18% | 14% | 16% | 18% | 18% | 16% | 17% | 15% | 14% | 15% | 11% | 14% |
| NC | 0% | 0% | 0% | 0% | 0% | 0% | 0% | 0% | 0% | 0% | 6% | 6% | 6% | 8% | 9% | 12% |
| ND | 0% | 0% | 0% | 0% | 0% | 0% | 0% | 0% | 0% | 0% | 0% | 0% | 0% | 0% | 2% | 2% |
| in low- (L_INC), lower-middle- (LM_INC) and upper-middle-income (UM_INC) countries | | | | | | | | | | | | | | | | |
| AD | 23% | 23% | 23% | 20% | 18% | 15% | 12% | 11% | 12% | 11% | 15% | 15% | 12% | 13% | 12% | 11% |
| RD | 25% | 24% | 34% | 42% | 45% | 53% | 55% | 60% | 57% | 61% | 55% | 49% | 47% | 48% | 46% | 42% |
| C | 42% | 45% | 40% | 36% | 35% | 30% | 30% | 27% | 30% | 27% | 30% | 33% | 39% | 34% | 36% | 41% |
| NC | 10% | 7% | 4% | 2% | 1% | 1% | 2% | 1% | 1% | 0% | 0% | 2% | 1% | 4% | 5% | 5% |
| ND | 0% | 1% | 0% | 0% | 1% | 1% | 1% | 0% | 0% | 1% | 0% | 0% | 1% | 0% | 1% | 1% |

**Table A2.** The share of the decoupling states in ten-year rolling windows from 1992–2017 for MF.

| Year | 2002 | 2003 | 2004 | 2005 | 20a06 | 2007 | 2008 | 2009 | 2010 | 2011 | 2012 | 2013 | 2014 | 2015 | 2016 | 2017 |
|------|------|------|------|------|-------|------|------|------|------|------|------|------|------|------|------|------|
| in all countries | | | | | | | | | | | | | | | | |
| AD | 35% | 33% | 27% | 20% | 14% | 11% | 10% | 13% | 11% | 12% | 9% | 9% | 13% | 16% | 18% | 18% |
| RD | 36% | 42% | 46% | 52% | 51% | 51% | 46% | 39% | 48% | 34% | 35% | 38% | 43% | 44% | 44% | 45% |
| C | 21% | 20% | 24% | 27% | 33% | 37% | 42% | 47% | 41% | 53% | 53% | 49% | 41% | 34% | 30% | 27% |
| NC | 7% | 6% | 3% | 1% | 1% | 1% | 2% | 1% | 1% | 1% | 1% | 1% | 2% | 3% | 4% | 4% |
| ND | 1% | 0% | 0% | 1% | 0% | 0% | 0% | 0% | 0% | 0% | 1% | 3% | 1% | 3% | 4% | 6% |
| in high-income (H_INC) countries | | | | | | | | | | | | | | | | |
| AD | 20% | 15% | 17% | 9% | 10% | 12% | 12% | 22% | 14% | 18% | 17% | 17% | 25% | 29% | 34% | 32% |
| RD | 66% | 56% | 54% | 66% | 69% | 64% | 49% | 38% | 50% | 43% | 33% | 46% | 43% | 38% | 40% | 40% |
| C | 14% | 29% | 29% | 26% | 21% | 24% | 40% | 40% | 36% | 39% | 44% | 31% | 25% | 25% | 15% | 14% |
| NC | 0% | 0% | 0% | 0% | 0% | 0% | 0% | 0% | 0% | 0% | 2% | 2% | 6% | 8% | 9% | 8% |
| ND | 0% | 0% | 0% | 0% | 0% | 0% | 0% | 0% | 0% | 0% | 4% | 4% | 0% | 0% | 2% | 6% |
| in low- (L_INC), lower-middle- (LM_INC) and upper-middle-income (UM_INC) countries | | | | | | | | | | | | | | | | |
| AD | 41% | 38% | 30% | 24% | 16% | 10% | 9% | 9% | 9% | 9% | 5% | 5% | 6% | 10% | 10% | 11% |
| RD | 26% | 37% | 43% | 47% | 44% | 45% | 45% | 40% | 46% | 30% | 37% | 34% | 42% | 47% | 46% | 48% |
| C | 23% | 17% | 23% | 27% | 38% | 42% | 43% | 50% | 43% | 60% | 58% | 58% | 50% | 39% | 38% | 34% |
| NC | 9% | 7% | 4% | 1% | 2% | 2% | 3% | 1% | 1% | 1% | 0% | 0% | 0% | 0% | 1% | 1% |
| ND | 1% | 0% | 0% | 1% | 0% | 0% | 0% | 0% | 0% | 0% | 0% | 2% | 2% | 4% | 5% | 5% |

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
