# Peer review of "Decoupling Economic Growth from Fossil Fuel Use—Evidence from 141 Countries in the 25-Year Perspective"

_energies, doi:10.3390/en13246671_

Round 1
Reviewer 1 Report
Decoupling economic growth from fossil fuel use – evidence from 141 countries in the 25-year perspective
The authors offer an in-depth analysis of decoupling economic growth from fossil fuel use in 141 countries with different levels of development over the last 25 years.
My points on this paper:
- In this paper, the authors make some interesting points on decoupling economic growth from fossil fuel use in 141 countries over the past 25 years.
- It is also obvious that countries that natural resources dependent will have difficulty to decouple easily.
- Issues raised with decoupling are related to the resource curse literature but the author(s) have said nothing about the resource curse thesis.
- Methods – data – how and why these countries are selected.
- What this kind of studies does is to musk the country-specific circumstances, which has been the challenge of the resource curse literature.
- There is the need to read through the paper thoroughly for editing. Some parts of the paper are poorly written, with wrong punctuations and capitalizations. The authors have to work on the abstract too since it captures the summary of the paper.
Author Response
C1: In this paper, the authors make some interesting points on decoupling economic growth from fossil fuel use in 141 countries over the past 25 years.
We would like to thank you for appreciating our work as well as for the comments, which we find very helpful in improving the manuscript. Below we provide a point-by-point response to the issues raised in the review.
C2: It is also obvious that countries that natural resources dependent will have difficulty to decouple easily.
We agree that that countries dependent on natural resources will have difficulty to decouple easily, and our research confirms this thesis.
C3: Issues raised with decoupling are related to the resource curse literature but the author(s) have said nothing about the resource curse thesis.
We would like to thank you for pointing out that the decoupling issues are also linked to the resource curse hypothesis, which states that countries endowed with natural resources tend to have lower economic growth and worse development outcomes than countries with fewer natural resources.
This hypothesis suggests that in resource-rich countries, despite an increase in fossil fuel use, these countries still have a low- and lower-middle income. This means that in these countries, a coupling or negative decoupling should be observed (see Figure 2 - Five decoupling states in the Tapio classification). The resource curse hypothesis applies only to resource-rich countries, i.e. about 50 countries (see IMF), and about 60% of these countries are low- and lower-middle-income.
Since the aim of our paper is to examine the problem of decoupling economic growth from the fossil fuels use on a global scale and to assess the level of transition of each country to a more resource-efficient economy and towards sustainable development and a green economy, we believe that analysis of only resource-rich countries would limit our subject matter.
However, we are convinced that this is a very interesting issue concerning the assessment of the relationship between fossil fuel use and economic growth, given the two approaches to measuring fossil fuel use in resource-rich countries. Such analysis can certainly contribute to the discussion on the resource curse hypothesis by confirming it or not. So, we would like to thank you for your this suggestion.
C4: Methods – data – how and why these countries are selected.
The selection of countries is based on the availability of data, which we mention in line 207. Our aim was a global analysis, and that is why we tried to select as many countries as possible with the analysis period as long as possible. Unfortunately, before 1992, many countries lacked DMC data. Even by limiting the period to the years 1992-2017, we did not manage to collect data for all the countries, so, finally, the analysis covers 141 countries.
C5: What this kind of studies does is to musk the country-specific circumstances, which has been the challenge of the resource curse literature.
Indeed, the raw material curse introduces an important context to the study of the decoupling of economic growth from resource use. Two points seem particularly important. Firstly, the countries affected by the curse have negative economic growth, so they will be classified exclusively as negative decoupling or negative coupling. Secondly, these countries are exporters of fossil fuels, so it will be possible to point out the contrast between the decoupling classifications obtained with the DMC and the MF. We find the reviewer’s suggestion so pertinent that we are going to study it separately.
In this study we provide a comparison for a very broad group of countries, a small percentage of which may be affected by the resource curse. In recent years, only three countries – Venezuela, Angola or Russia – have been mentioned in this context. Thus it seems that a separate study will be more appropriate to comment on decoupling in this context. Nevertheless, in the introduction section we mention some papers that are devoted to decoupling analysis in the context of fossil fuel export or resources curse.
C6: There is the need to read through the paper thoroughly for editing. Some parts of the paper are poorly written, with wrong punctuations and capitalizations. The authors have to work on the abstract too since it captures the summary of the paper.
To address this comment, we asked a native English speaker to proofread the text. We hope that the revised version of the manuscript has been satisfactorily improved in terms of its linguistic quality.
As far as our abstract is concerned, we tried to fulfil the editor’s requirements. In particular, the abstract should include four parts: 1) Background, 2) Methods, 3) Results 4) Conclusion. However, following this suggestion, we have modified the summary to make it clearer.
Reviewer 2 Report
General comments
The authors propose, in the submitted paper energies-1018842, a paper about “Decoupling economic growth from fossil fuel use – 3 evidence from 141 countries in the 25-year 4 perspective”. In this paper, the study finds that the process of decoupling economic growth from fossil fuel use differs when different measures of fossil fuel use are considered. In particular, when the DMC indicator is considered, relative decoupling is observed in most cases. In the case of the MF indicator, the decoupling states of individual countries change more frequently. Finally, in highly developed countries absolute decoupling is frequently identified, although only when the DMC measurement is used. The paper can be of great interest to energies readers, I would suggest some improvements to improve its global quality:
- Include a nomenclature section, if possible
- Page 6: I would suggest avoiding the footnotes and include the information in the review section or in the references
- Figure 4: they are difficult to read and might be the same information that is shown in Figure 3. The information might be repetitive.
- Figure 5 and figure 6 should be improved
- Remark the novelty of the research
- Provide a flow chart for the global research
The reviewer.
Author Response
We would like to thank you for appreciating our work as well as for the comments, which we find very helpful in improving the manuscript. Below we provide a point-by-point response to the issues raised in the review.
C1: Include a nomenclature section, if possible
Due to the template of the manuscript recommended by “Energies”, we have decided to leave the abbreviations in the main text.
C2 Page 6: I would suggest avoiding the footnotes and include the information in the review section or in the references
The footnotes 5-8 on page 6 have been replaced by references. The content of the third footnote has been entered into the main text.
C3 Figure 4: they are difficult to read and might be the same information that is shown in Figure 3. The information might be repetitive.
The readability of Figure 4 has been improved, the data have been moved to Table A1 (Appendix).
C4: Figure 5 and figure 6 should be improved
Figure 5 and figure 6 have been improved.
C5: Remark the novelty of the research
In the Introduction section, we emphasize the three novelty aspects of our research (line 153). Also, we think that line 109 provides information on the novelty (This study aims to fill this gap by in-depth analysis of the decoupling of economic growth from fossil fuel use in 141 countries characterised by different levels of development over the last 25 years (1992–2017).)
C6: Provide a flow chart for the global research
We have provided a flow chart for our global research. This flow chart is presented in Figure 1. Adding figure 1 caused changes in the numbering of consecutive figures.
Reviewer 3 Report
Title: Decoupling economic growth from fossil fuel use – evidence from 141 countries in the 25-year perspective
Manuscript ID: energies-1018842-
This study analyses the process of decoupling economic growth from fossil fuels use in 141 countries with different levels of development over the last 25 years. The Tapio decoupling approach and two measures of fossil fuels use are applied. Acknowledging the contributions of the manuscript, I do not have problems of this manuscript.
Authors need to pay attention to format specification. For example, on page 3, line 133. It would be better if the author can well explain the meaning of characters in the equations.
Author Response
We would like to thank you for appreciating our work as well as for the comments, which we find very helpful in improving the manuscript. Below we provide a point-by-point response to the issues raised in the review.
C1 Authors need to pay attention to format specification. For example, on page 3, line 133.
The format specification has been improved, especially on page 3, line 133.
C2 It would be better if the author can well explain the meaning of characters in the equations.
The meaning of characters in equation 1 has been improved.
Round 2
Reviewer 1 Report
I think my issues have been resolved and if the editors are comfortable, the paper can proceed.
Reviewer 2 Report
The paper is greatly improved and the reviewer's comments are addressed.
Sincerely, the reviewer.